# Simultaneous Measurement of Group Refractive Index Dispersion and Thickness of Fused Silica Using a Scanning White Light Interferometer

**DOI:** 10.3390/s24010017

**Published:** 2023-12-19

**Authors:** Heesu Lee, Seungjin Hwang, Hong Jin Kong, Kyung Hee Hong, Tae Jun Yu

**Affiliations:** 1Department of Computer Science and Electrical Engineering, Handong Global University, Pohang 37554, Republic of Korea; heesu.lee@handong.ac.kr; 2HIL Lab Inc., Pohang 37563, Republic of Korea; seungjin.hwang@hillab.co.kr; 3Department of Physics, Korea Advanced Institute of Science, Daejeon 34141, Republic of Korea; konghj@kaist.ac.kr; 4Department of Advanced Convergence, Handong Global University, Pohang 37554, Republic of Korea; khhong@handong.edu

**Keywords:** white light interferometer, scanning white light interferometer, group refractive index dispersion, thickness, fused silica

## Abstract

In this study, we simultaneously measured the group refractive index dispersion and thickness of fused silica using a scanning white light interferometer on a spectral range from 800 to 1050 nm. A delay error correction was performed using a He-Ne laser. The accuracy of the measured group refractive index dispersion of fused silica, when compared to the temperature-dependent Sellmeier equation, is within 4 × 10^−5^.

## 1. Introduction

The group refractive index is the ratio of the group velocity of light in a medium to the speed of light in a vacuum. The group refractive index can be used to determine the speed of a pulse propagating in a particular medium. The formula for the group refractive index can be expressed as
(1)$$N=n-\lambda \frac{dn}{d\lambda }$$
where N is the group refractive index, n is the refractive index, and λ is the wavelength.

The dispersion of the group refractive index is the variation in the group refractive index with wavelength. Group refractive index dispersion can be used to calculate group velocity dispersion (GVD) or group delay dispersion (GDD).

The group velocity in optical fibers used in ultra-fast optical communications is very important. GDD affects the propagation of optical pulses transmitted through optical fibers, and also affects the transmission capacity of the fiber. Therefore, it is important to know the GVD of the fiber [1]. In addition to basic science, group velocity is also very important in ultra-fast lasers, which are widely used in industrial applications. GDD plays an important role, not only in optical fiber communications, but also in femtosecond laser generation. If the GDD of the optical system of a femtosecond oscillator is accurately known, the shape of the generated femtosecond pulse can be simulated, and a compensation design can be made to prevent the deformation of the pulse [2]. Chirped-pulse amplification (CPA) is a technique used to amplify GW laser pulses to TW or PW levels [3,4,5,6,7]. GDD is also important in CPA. This technique generates ultra-high-power lasers without damaging the optical system by stretching the seed beam in a pulse stretcher, amplifying it, and then compressing it [8]. It is important to know the GDD accurately to adjust the length of the femtosecond laser in the process of stretching and compressing the pulse [9].

Many studies have been conducted to measure group refractive index dispersion. Delbarre et al. used a white light spectral interferometer to measure the group refractive index of 540–660 nm with an accuracy of 9 × 10^−4^; they measured the thickness at the same time [10]. This method possesses a disadvantage in that one is required to perform a measurement three times to measure 540–660 nm when using a spectrometer that can only measure a 40 nm range at a time. In addition, the accuracy of the group refractive index measurement was as low as a 10^−3^ level. Hlubina used a spectral domain white light interferometer to measure group refractive index dispersion in the 500–900 nm range [11]. The accuracy was 7 × 10^−5^, and the thickness of the sample was measured at the same time. However, it is necessary to measure the group refractive index dispersion and the displacement of the mirror at each wavelength point; therefore, the number of measurements used previously was too large. Arosa et al. used a spectrally resolved white light interferometer to measure the group refractive index of a wide band of 400–1550 nm with an accuracy of 10^−4^ [12]. The sample thickness should be measured separately.

A review of previous studies shows that the simultaneous measurement of sample thickness is required to achieve a 10^−5^ level [11,12]. This is because the angle of incidence of light changes depending on the alignment state of the sample, and the group refractive index measured accordingly also changes, resulting in measurement errors [13]. Therefore, it is essential to measure the sample thickness simultaneously with the group refractive index to accurately measure the group refractive index.

Most previous studies have been conducted using spectral methods. Spectral methods measure spectral interferograms, and an equalization wavelength is generated near the optical path difference of 0 in both arms. In this vicinity, the refractive index dispersion is measured throughout the period of the spectral fringe [14,15]. The spectral method measures the refractive index by using the periodicity of the fringe. However, it has a disadvantage in that it is difficult to resolve issues where the measurement medium is thick or dispersive, which limits the measurement sample [16]. Further, the spectral method requires an accurate calibration of the CCD array or spectrometer, which makes it difficult to configure and manage the setup. In addition, the spectral resolution is limited by the spectrometer or CCD specifications, which results in poor resolution [17].

The scanning method used in this study can compensate for the disadvantages of the existing methods. It possesses an advantage in that it can measure more dispersive or thicker samples than the spectral method by simply increasing the scanning distance. Unlike the spectral method, the measurement sensor is a photodiode or avalanche photodiode, so calibration is not required, making it relatively easy to manage. In addition, the spectral resolution improves in proportion to the scanning distance, and the resolution is superior to that of the spectral method. The scanning method measures a temporal interferogram in the time domain. It measures the delay of the optical pulse passing through the sample. The interferogram can be Fourier-transformed to obtain the entire spectrum, and the dispersion can be obtained by differentiating the spectral phase obtained by the Fourier transformation. The properties of the Fourier transform method and the dispersive method are summarized in Table 1 on the following page.

Due to the change in the angle of incidence of the sample depending on the measurement, it is necessary to simultaneously measure the thickness to compensate for it. By setting the scanning distance long enough, the internal reflection signal of the interferogram can be measured at the same time as the sample interferogram. The thickness of the sample can be calculated by the Fourier transformation of this signal.

In this study, we simultaneously measured the group refractive index dispersion and the thickness of fused silica using a scanning white light interferometer on a spectral range from 800 to 1050 nm. We present a detailed analysis of the measured data from the scanning white light interferometer.

## 2. Scanning White Light Interferometer Experimental Setup

The interferometer used in this study is a Michelson type. Figure 1 displays the experimental setup of the measurement. The optical paths were aligned to facilitate the alignment of the laser and the white light. A white light source was used as a light source for the group refractive index measurement, and a He-Ne laser was used for the scanning delay measurement. Scanning is performed by moving M4 using the voice coil motor (VCM) stage of the lower arm of the interferometer. The sample is mounted on the right arm of the Michelson interferometer and passes through twice due to the Michelson type. The He-Ne interferogram was measured by a photodetector (PD), and the white light interferogram was measured by an avalanche photodiode (APD).

The He-Ne laser is a stabilized He-Ne laser from SIOS (Ilmenau, Germany), model SL-04A. The vacuum wavelength is 632.9911242 ± 1.2 × 10^−6^ nm. The white light source is a stabilized white light source from Thorlabs (Newton, NJ, USA), model SLS201L. The wavelength is 300–2600 nm, and the output stability is below 0.05%. The VCM stage is a VCM stage from Technohands (Yokohama, Japan), model VXH11-10. The PD is from Thorlabs, model DET10A2. The APD is fabricated using an APD element from Hamamatsu (Hamamatsu, Japan), model S11519-10.

There are two types of mirrors used in the interferometer: silver mirrors (M) and dichroic mirrors (DM). The silver mirror is a Thorlabs PF10-03-P01, and the dichroic mirror is a Thorlabs DMSP650 with a cutoff wavelength of 650 nm. The beam splitter is a Thorlabs BS014 with a wavelength range of 700–1100 nm. A 700 nm longpass filter is used to block the He-Ne laser from being measured by the APD. The lens is a Thorlabs achromatic doublet, AC254-050-B-ML, with a focal length of 50 mm and an AR coating band of 650–1050 nm.

The sample used in the experiment is fused silica, with a thickness of 3 mm and a diameter of 25.4 mm. The flatness is λ/10, and there is no coating.

The VCM stage performs 5 consecutive scans, each with a round-trip travel distance of 1 cm. The oscilloscope’s sampling rate is 1 MS/s for PD and APD measurements, and the total measurement time is 10 s.

## 3. Analysis

The scanning speed of the VCM used to measure the temporal interferogram is not uniform; therefore, it needs to be corrected. The device used to correct this is a stabilized He-Ne laser. The delay of one period of the He-Ne laser can be calculated using Equations (2) and (3) with a vacuum wavelength and an air refractive index. It corresponds to about 2.11 fs, and the measurement data in the time domain can be converted to the delay domain based on the He-Ne interferogram. Figure 2 and Figure 3 relate to the delay error correction process. The vacuum wavelength of the He-Ne laser and the delay of one period can be expressed as
(2)$$\lambda_{\mathrm{vac}}≅632.991\;\mathrm{nm}$$
(3)$$\tau_{1}\equiv \frac{\lambda_{\mathrm{vac}}}{c}≅2.11143\;\mathrm{fs}$$
where $\lambda_{\mathrm{vac}}$ is the vacuum wavelength and $\tau_{1}$ is the delay of one period of the He-Ne laser.

Furthermore, the number of He-Ne laser fringes can be counted according to the scanning direction of the VCM. Since the He-Ne laser has a fixed wavelength, the distance of each scan can be accurately determined by counting the zero crossings of the fringe. By interpolating the white light interferogram based on the He-Ne laser fringe, the unevenness in the time domain due to the scanning speed of the VCM can be removed. Since the scanning direction and distance of the VCM are known, the scanning data can be parameterized based on the zero crossings. This can be used to multiplex multiple consecutive scanning data to reduce noise. The He-Ne laser and white light interferogram of the time domain and the delay domain can be expressed as
(4)$$I\left(t\right)=I_{0}\left(t\right)+V\left(t\right)cos\phi \left(t\right)$$
(5)$$I\left(\tau \right)=I_{0}\left(\tau \right)+V\left(\tau \right)cos\phi \left(\tau \right)$$
where $I\left(t\right)$ is the intensity of the interferogram in the time domain, $I\left(\tau \right)$ is the intensity of the interferogram in the delay domain, $I_{0}$ is the intensity of the light source without interference, V is the envelope function of the interferogram, and ϕ is the phase. Equation (4) can be transformed to Equation (5) by delay error correction. As the VCM scanning distance increases from 0 to 1 cm, the offset of the white light interference signal decreases gradually. This can be seen in Figure 2c,d.

The white light interferogram of the fused silica sample in the delay domain, calculated via the delay error correction process, and the blank white light interferogram without the sample are both Fourier-transformed. This results in a complex spectrum in the wavelength domain. The spectrum and phase of the sample and the blank interferogram can be expressed as
(6)$$S_{Samp}\;\left(\omega \right)=\int_{-\infty }^{\infty }I_{Samp}\left(\tau \right)e^{i\omega \tau }d\tau$$
(7)$$S_{Blank}\;\left(\omega \right)=\int_{-\infty }^{\infty }I_{Blank}\left(\tau \right)e^{i\omega \tau }d\tau$$
(8)$$\phi_{Samp}\left(\omega \right)=arctan\left(\frac{Im\left\{S_{samp}\left(\omega \right)\right\}}{Re\left\{S_{samp}\left(\omega \right)\right\}}\right)$$
(9)$$\phi_{Blank}\left(\omega \right)=arctan\left(\frac{Im\left\{S_{Blank}\left(\omega \right)\right\}}{Re\left\{S_{Blank}\left(\omega \right)\right\}}\right)$$
where $S_{Samp}$ is the spectrum of the fused silica interferogram, $S_{Blank}$ is the spectrum of the interferogram without a sample, $\phi_{Samp}$ is the phase of the fused silica interferogram, and $\phi_{Blank}$ is the phase of the interferogram without a sample.

The phase can be obtained using Equations (6) and (7). However, this is a wrapped phase in the range of −π to π, so the phase can be obtained by unwrapping it using Equations (8) and (9). The intensity can be obtained by taking the absolute value of the complex spectrum.

Figure 4 shows the white light spectrum calculation process. In Figure 4a, an internal reflection interference signal can be observed. The delay between the main interference signal and the internal reflection signal is consistent with the round-trip delay of the internal reflection in the sample. This can be calculated using Equation (10). The delay between the main interference signal shown in Figure 4a,c is consistent with the delay by inserting the sample in one arm. The blank interference is shifted by inserting the fused silica sample in the interferometer’s arm. The delay difference can be calculated with Equation (11):(10)$$\tau_{1}=\frac{N_{fs}L}{2c}$$
(11)$$\tau_{2}=\frac{(N_{fs}-N_{air})L}{2c}$$

Via Fourier transformation of the wavelength absolute spectrum of fused silica, the internal reflection signal can be found in the delay domain. The reflection signal and thickness of the fused silica sample can be expressed as
(12)$$\bar{\tau }=\frac{\int I\left(\tau \right)\tau d\tau }{\int I\left(\tau \right)d\tau }$$
(13)$$L=\frac{\bar{\tau }c\;}{2n_{group}\left(\bar{\lambda }\right)}$$
where $\bar{\tau }$ is the center of mass of the sample reflection signal and L is the thickness of the sample. The thickness of the sample can be calculated using the known group refractive index of the wavelength spectrum, the center of mass of the fused silica, and the center of mass of the reflection signal using Equations (12) and (13).

By calculating the phases of the fused silica sample and blank and subtracting the blank phase from the fused silica phase, the phase delayed by fused silica can be calculated and expressed as
(14)$$\phi_{\Delta }=\phi_{Samp}\left(\lambda \right)-\phi_{Blank}\left(\lambda \right)\;=\phi_{Fused\;silica}\left(\lambda \right)-\phi_{air}\left(\lambda \right)$$
where $\phi_{\Delta }$ is the phase delayed by fused silica. It can be expressed by the following equation with the refractive index of the fused silica and the air sample thickness. The theoretical equation of $\phi_{\Delta }$ can be expressed as
(15)$$\phi_{\Delta }\left(\lambda \right)=\arctan\left(\frac{Im\left(S\left(\lambda \right)\right)}{Re\left(S\left(\lambda \right)\right)}\right)+2m\pi \;=4\pi L(n_{FS}\left(\lambda \right)-n_{air}\left(\lambda \right))/\lambda$$
where $n_{FS}$ is the refractive index of the fused silica and $n_{air}$ is the refractive index of the air.

By differentiating the $\phi_{\Delta }$, the group refractive index of fused silica can be calculated. The equations for this can be expressed as
(16)$$\frac{\partial \phi_{\Delta }}{\partial \lambda }=-\frac{4\pi L}{\lambda^{2}}\left(n_{FS}\left(\lambda \right)+\lambda \frac{\partial n_{FS}}{\partial \lambda }-n_{air}\left(\lambda \right)-\lambda \frac{\partial n_{air}}{\partial \lambda }\right)$$
(17)$${\left.\frac{\partial \phi_{\Delta }}{\partial \lambda }\right|}_{\lambda =\lambda_{0}}=-\frac{4\pi L}{\lambda_{0}^{2}}(N_{FS}\left(\lambda_{0}\right)-N_{air}\left(\lambda_{0}\right)$$
(18)$$N_{FS}\left(\lambda_{0}\right)=-\frac{\lambda_{0}^{2}}{4\pi L}{\left.\frac{\partial \phi_{\Delta }}{\partial \lambda }\right|}_{\lambda =\lambda_{0}}+N_{air}\left(\lambda_{0}\right)$$
where $N_{FS}$ is the group refractive index of the fused silica and $N_{air}$ is the group refractive index of the air.

With the derivative of $\phi_{\Delta }$ and the group refractive index of the air, the group refractive index of the fused silica can be calculated using Equation (18).

By subtracting the measured group refractive index of the spectrum’s center of mass from this, the group refractive index dispersion can be calculated. The equations for the calculation of the group refractive index dispersion can be expressed as
(19)$$\Delta N_{M}\left(\lambda \right)\equiv N_{FS\;measured}\left(\lambda \right)-N_{FS\;measured}\left(\bar{\lambda }\right)$$
(20)$$\Delta N_{S}\left(\lambda \right)\equiv N_{FS\;Sellmeier}\left(\lambda \right)-N_{FS\;Sellmeier}\left(\bar{\lambda }\right)$$
(21)$$\Delta N_{error}\left(\lambda \right)\equiv \Delta N_{M}\left(\lambda \right)-\Delta N_{S}\left(\lambda \right)$$
where $\Delta N_{M}\left(\lambda \right)$ is the measured group refractive index dispersion, $\Delta N_{S}\left(\lambda \right)$ is the theoretical group refractive index dispersion, and $\Delta N_{error}\left(\lambda \right)$ is the group refractive index dispersion error.

## 4. Experimental Result and Discussion

The thickness was calculated via the Fourier transformation of the measured fused silica spectrum [18]. The thickness was measured 10 times, and these measurements are shown in Table 2. The standard deviation of the thickness measurement calculated from the 10 measurement results is 70 nm, and the average value is 3.05009 mm. The thickness calculation process and thickness measurement results are shown in Figure 5 and Figure 6.

The results of the fused silica group refractive index dispersion measurement performed in this study are shown in Figure 7. The standard deviation of the group refractive index dispersion of (d) in Figure 7 is within 1.3 × 10^−5^, as shown in Figure 8. The measured dispersion values were compared with the theoretical values calculated by the temperature-dependent Sellmeier equation [19]. The measured fused silica dispersion accuracy was within 4 × 10^−5^ compared to the theoretical value. The group refractive index dispersion is calculated based on $\bar{\lambda }$, and converges at $\bar{\lambda }$. The repeatability of the group refractive index dispersion can be determined by the STD shown in Figure 8.

## 5. Conclusion and Future Study

We simultaneously measured the group refractive index dispersion and thickness of fused silica using a scanning white light interferometer. The delay error correction was performed using the He-Ne laser. The white light spectrum of the measured fused silica was Fourier-transformed to obtain a reflection signal, which was used to calculate the thickness of the fused silica sample. The standard deviation of the thickness measurement was 70 nm, and the average value was 3.05009 mm. The group refractive index was calculated using the derivative of the measured fused silica phase. The accuracy in the measured group refractive index dispersion of the fused silica, when compared to the temperature-dependent Sellmeier equation, was within 4 × 10^−5^.

## Figures and Tables

**Figure 1 sensors-24-00017-f001:**
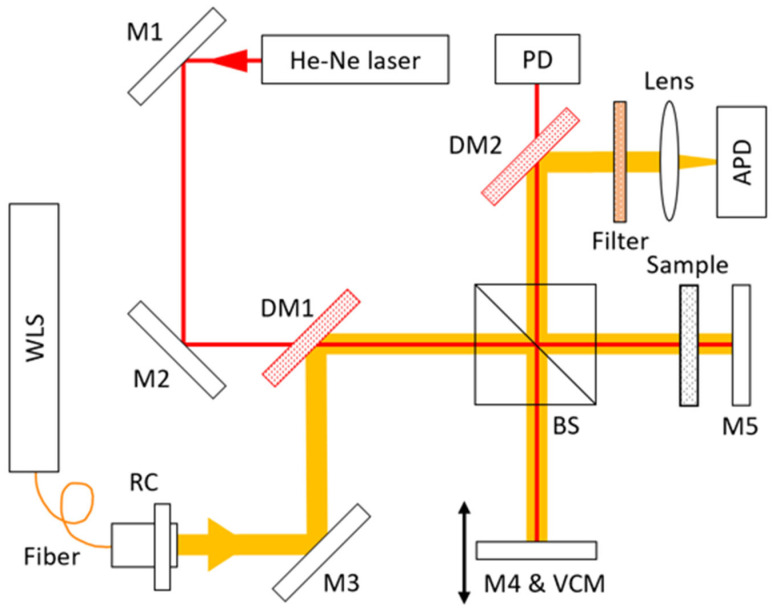
A schematic diagram of the interferometer experimental setup (M: mirror, DM: dichroic mirror, BS: beam splitter, WLS: white light source, RC: reflective collimator, VCM: voice coil motor stage, PD: photodetector, and APD: avalanche photodetector). Yellow line is optical path of white light. Red line is optical path of He-Ne laser. Arrows show the direction of white light and He-Ne laser.

**Figure 2 sensors-24-00017-f002:**
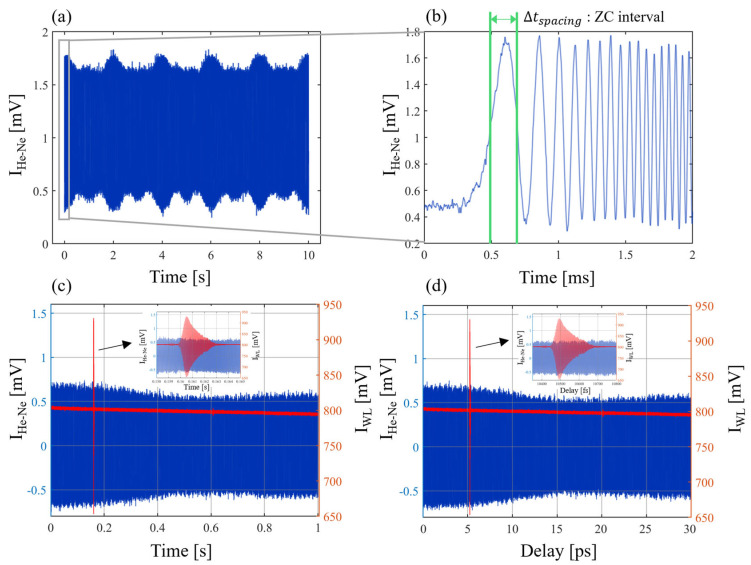
The delay error correction process. The (**a**) He-Ne interferogram, (**b**) zoomed-in graph of the He-Ne interferogram and an explanation of the zero-crossing (ZC) interval, (**c**) white light and He-Ne interferogram in the time domain, and (**d**) white light and He-Ne interferogram in the delay domain. $I_{He-Ne}$ is the He-Ne interferogram and $I_{WL}$ is the white light interferogram.

**Figure 3 sensors-24-00017-f003:**
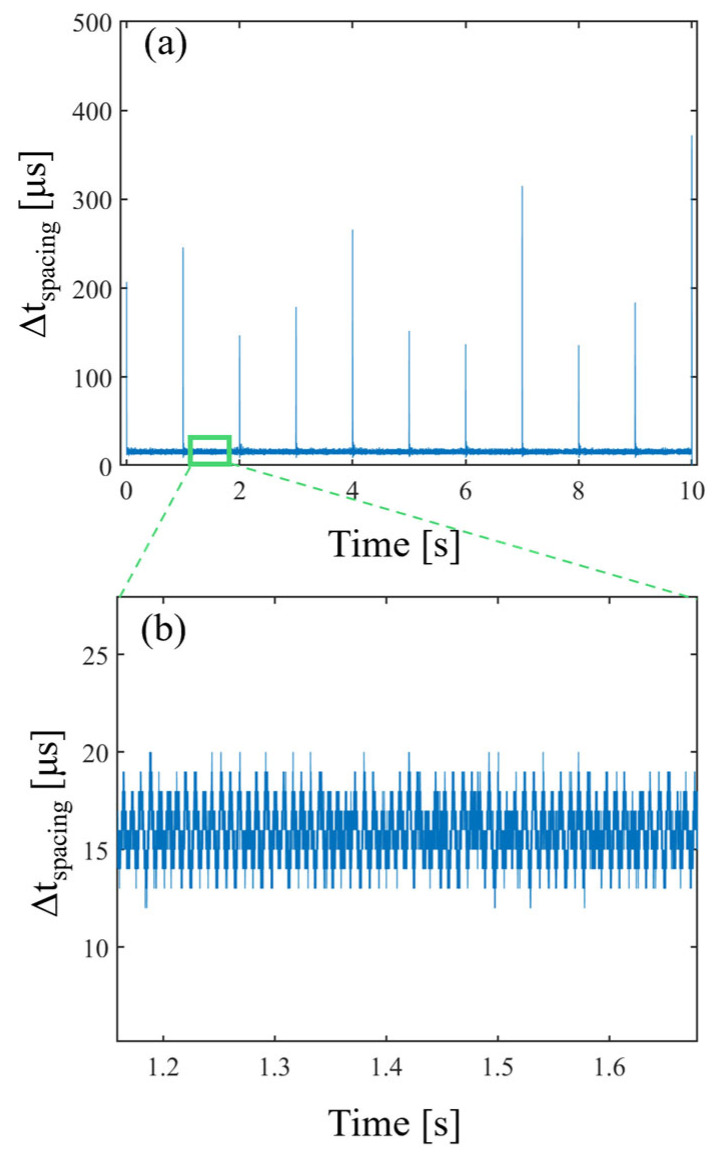
The unevenness of a zero-crossing time interval. The (**a**) zero-crossing time interval and (**b**) zoomed-in graph of the zero-crossing time interval.

**Figure 4 sensors-24-00017-f004:**
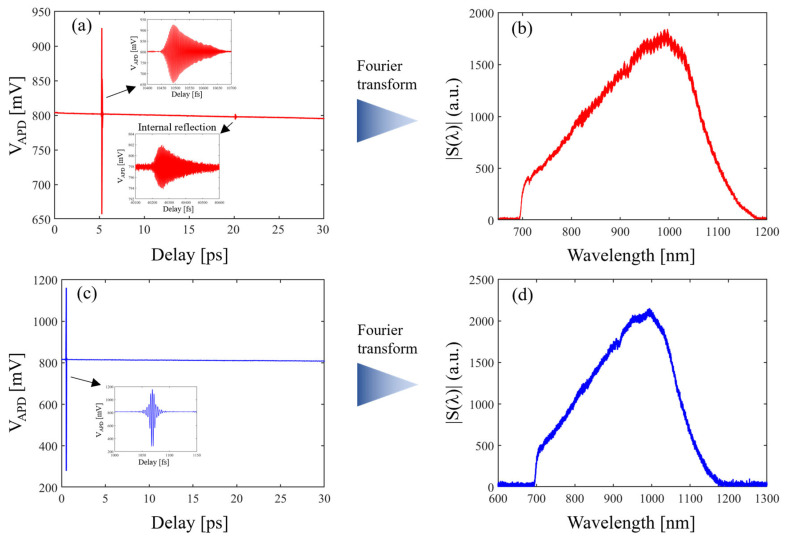
The wavelength spectrum calculation of the white light interferogram. The (**a**) fused silica interferogram in the delay domain, (**b**) fused silica spectrum, (**c**) blank interferogram in the delay domain, and (**d**) blank spectrum.

**Figure 5 sensors-24-00017-f005:**
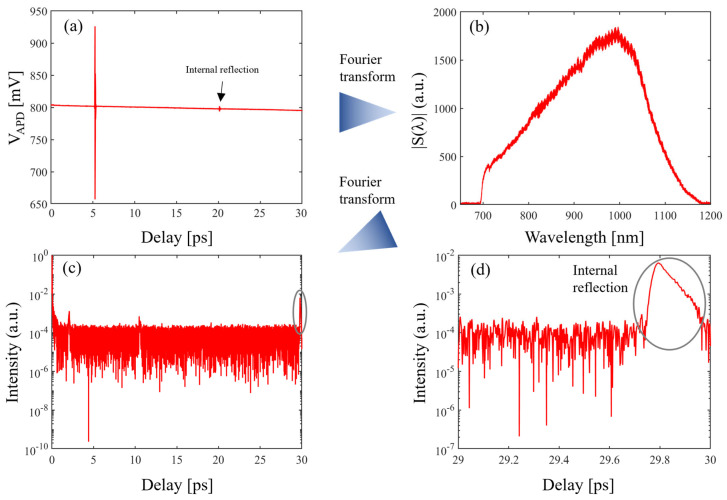
The fused silica sample internal reflection calculation process. Sample thickness is calculated using an internal reflection and a known group refractive index. The (**a**) fused silica interferogram in the delay domain, (**b**) fused silica spectrum, (**c**) Fourier transformation of the fused silica spectrum, and (**d**) zoomed-in graph of the Fourier transformation of the fused silica spectrum.

**Figure 6 sensors-24-00017-f006:**
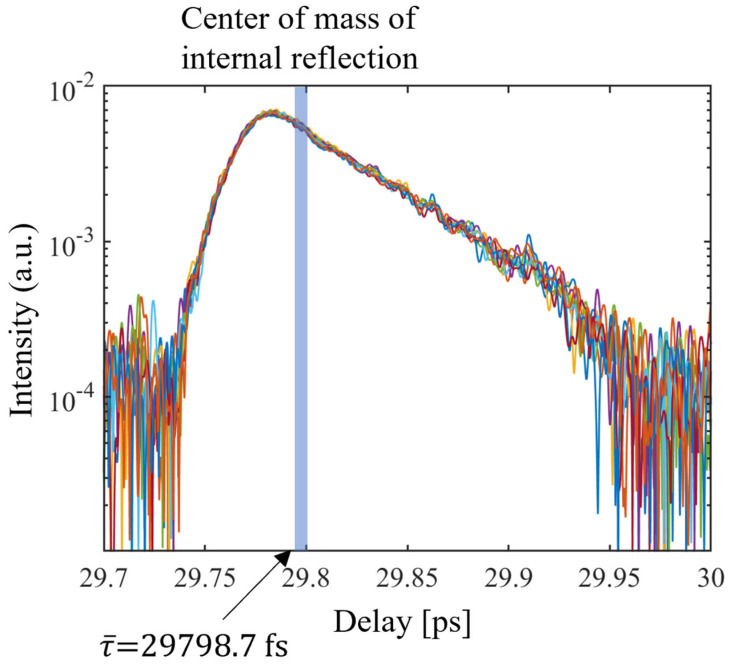
The fused silica sample internal reflection (measured 10 times).

**Figure 7 sensors-24-00017-f007:**
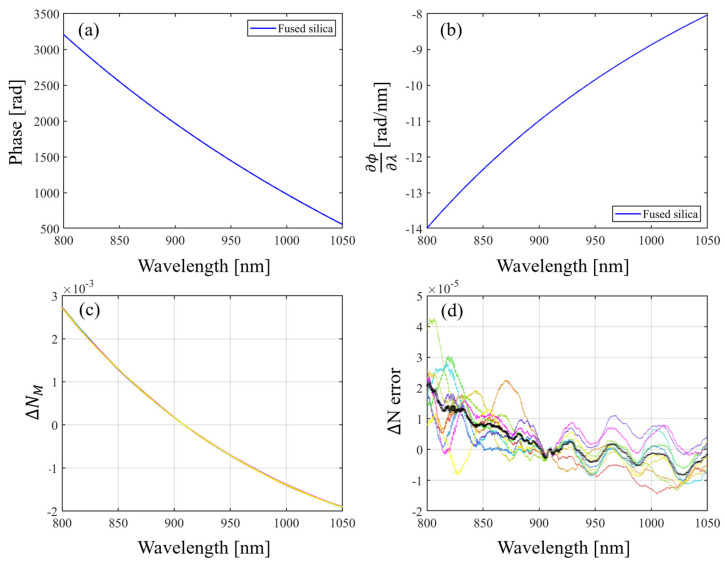
The group refractive index measurement process of the fused silica. The (**a**) measured phase of the fused silica, (**b**) differentiation of the phase of fused silica, (**c**) measured group refractive index dispersion (measured 10 times), and (**d**) group refractive index dispersion error (measured 10 times, the black bold line represents the average of the 10 measurements). Measured group refractive index dispersion and group refractive index dispersion error can be calculated using Equations (19)–(21).

**Figure 8 sensors-24-00017-f008:**
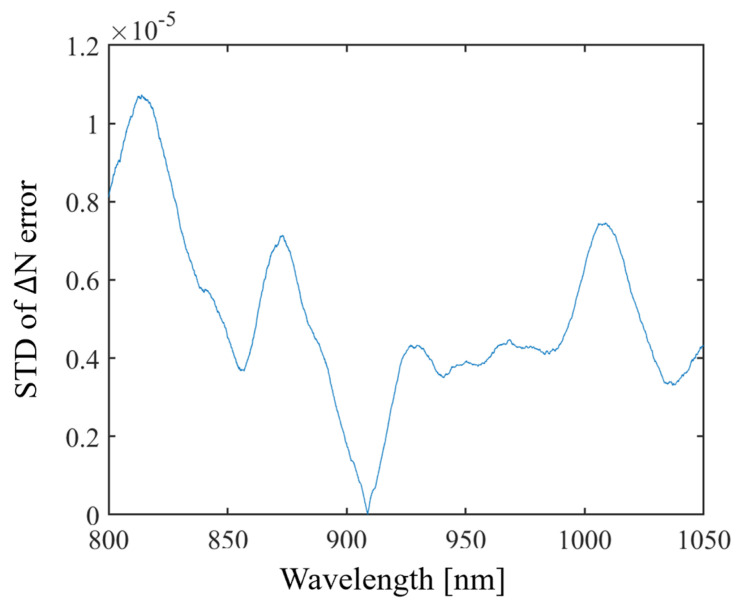
The standard deviation of the group refractive index dispersion accuracy of (d) in Figure 7.

**Table 1 sensors-24-00017-t001:** A comparison of the properties of the Fourier transform method and the dispersive method.

	Fourier Transform Method	Dispersive Method
Measurement domain	Temporal	Spectral
Signal to Noise ratio	good	bad
Throughput	good	bad
Wavelength calibration precision	good	bad
Tradeoff (resolution vs. throughput)	no	yes
Maintenance (calibration)	Unnecessary	Necessary
Data length	long	short

**Table 2 sensors-24-00017-t002:** The measured fused silica sample thickness (measured 10 times).

Thickness (mm)			
#1	3.05019	#6	3.05001
#2	3.05007	#7	3.05006
#3	3.05018	#8	3.05001
#4	3.05017	#9	3.05003
#5	3.05008	#10	3.05011
Mean (L)		3.05009 mm	
Standard deviation (σ)		70 nm	

## Data Availability

The data presented in this study are available on request from the corresponding author. The data are not publicly available due to secrecy.

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
