# Peer review of "Simultaneous Measurement of Group Refractive Index Dispersion and Thickness of Fused Silica Using a Scanning White Light Interferometer"

_sensors, 2023, doi:10.3390/s24010017_

Round 1

Reviewer 1 Report

Comments and Suggestions for Authors

The manuscript under discussion is devoted to the experimental investigation of the group refractive index dispersion in fused silica in the near-infrared spectral range. The authors present a usage of experimental technique based on Michelson interferometer with original options, which allows to provide measurements in time domain as well to simultaneously measure the thickness of studied sample. As a result, authors obtained spectral dependence of the group refractive index in the wavelength range of 800-1050 nm, however the provided spectral studies can be extended with the usage of longwave photodetectors and appropriate optic elements.

The manuscript is well-written and can be published after minor revision:

1. The white-light interferogram, presented in Fig. 2c and 2d, shows a slight incline (the amplitude is about 820mV  for t=0s and 805 mV for t=1s). What is the reason for such a incline?

2. The WL interferogram presented in fig.4a is consisted of slight interference signal at the time t=40ps in addition to main part, located close to the t=10000s. What is the reason for such an additional interference at the t=40ps? Why it is not observed in the blank interferogram, presented in fig. 4c?

3. Why interference maximum is shifted in time scale for the WL and blank interferograms, presented in Fig. 4a and 4c, respectively? 

Author Response

1. Summary

Thank you very much for taking the time to review this manuscript. Please find the detailed responses below and the corresponding revisions/corrections highlighted/in track changes in the re-submitted files. Data of figure 2, 3, 4, 5 were changed because mis-selected data were used in last submitted manuscript. The measurement results and conclusions are the same because the only mistake was in the selection of the figures.

2. Point-by-point response to Comments and Suggestions for Authors

The manuscript under discussion is devoted to the experimental investigation of the group refractive index dispersion in fused silica in the near-infrared spectral range. The authors present a usage of experimental technique based on Michelson interferometer with original options, which allows to provide measurements in time domain as well to simultaneously measure the thickness of studied sample. As a result, authors obtained spectral dependence of the group refractive index in the wavelength range of 800-1050 nm, however the provided spectral studies can be extended with the usage of longwave photodetectors and appropriate optic elements.

The manuscript is well-written and can be published after minor revision:

Comments 1: The white-light interferogram, presented in Fig. 2c and 2d, shows a slight incline (the amplitude is about 820mV for t=0s and 805 mV for t=1s). What is the reason for such a incline?

Response 1: Thank you for the comment. We have added explanation about this in manuscript. As the VCM scanning distance increases from 0 to 1 cm, offset of white light interference signal decreases gradually. This explanation is added in line number 158 – 160. And corrected figure 2. Added grid to distinguish the incline.

Comments 2: The WL interferogram presented in fig.4a is consisted of slight interference signal at the time t=40ps in addition to main part, located close to the t=10000s. What is the reason for such an additional interference at the t=40ps? Why it is not observed in the blank interferogram, presented in fig. 4c?

Response 2: Thank you for the comment. Explanation about white light interferogram was not enough. We have added explanation and equation (10) about this in manuscript. The additional interference is generated by internal reflection of sample. So, it can’t exist in blank interferogram. The delay between main interference signal and internal reflection signal is consistent with the round-trip delay of the internal reflection in the sample. The explanation is added in line number 176-179. And corrected figure 2 to know internal reflection interference.

Comments 3: Why interference maximum is shifted in time scale for the WL and blank interferograms, presented in Fig. 4a and 4c, respectively?

Response 3: Thank you for the comment. Explanation about white light interferogram was not enough. We have added explanation and equation (11) about this in manuscript. The interference is shifted by insertion of fused silica sample in one arm of the interferometer. The explanation is added in line number 179-182.

Reviewer 2 Report

Comments and Suggestions for Authors The authors used a scanning white light interferometer to measure the refractive indexdispersion and thickness of fused silica. This paper can be accepted after some few minor comments: 1) The authors need to present the different methods of interferometry used in the determination of these properties in a tabular form for the benefit of the readers 2) More statistical analysis of their results must also be included aside from the standard deviation being presented. Figure 7d, statistical analysis should also be included, although they presented somewhat in Figure 8 but this does not represent if the measurements are in relation to each other. 3) Explain the sudden blips in their interferogram in Figures 2, 4, and 5?

Author Response

1. Summary

Thank you very much for taking the time to review this manuscript. Please find the detailed responses below and the corresponding revisions/corrections highlighted/in track changes in the re-submitted files. Data of figure 2, 3, 4, 5 were changed because mis-selected data were used in last submitted manuscript. The measurement results and conclusions are the same because the only mistake was in the selection of the figures.

2. Point-by-point response to Comments and Suggestions for Authors

The authors used a scanning white light interferometer to measure the refractive index dispersion and thickness of fused silica. This paper can be accepted after some few minor comments: The manuscript is well-written and can be published after minor revision:

Comments 1: The authors need to present the different methods of interferometry used in the determination of these properties in a tabular form for the benefit of the readers.

Response 1: Thank you for the comment. We also think this would be helpful to the readers. We have added table explaining the properties of measurement methods. The table is in line number 97.

Comments 2: More statistical analysis of their results must also be included aside from the standard deviation being presented. Figure 7d, statistical analysis should also be included, although they presented somewhat in Figure 8 but this does not represent if the measurements are in relation to each other.

Response 2: Thank you for the comment. Explanation about figure 8 was not enough. We corrected 236th line of manuscript to make it easier to understand the connection between 7d and 8. And added explanation of statistical meaning of standard deviation in line number 239 – 241. And corrected figure 7d and 8 for intuition. Black bold line of figure 7d is average of measurements. And figure 8 is for the standard deviation.

Comments 3: Explain the sudden blips in their interferogram in Figures 2, 4, and 5?

Response 3: Thank you for the comment. We added inset to show detailed interference signals in figure 2, 4. The inset of figure 5 is omitted because it is the same as figure 4. Sudden blips are optical interference phenomena that occur when the optical paths are almost aligned (within the coherence length of white light) as the VCM stage is scanned. The red line is the interference signal measured with the fused silica sample inserted, and the blue line is the interference signal without the fused silica sample.